# The Role of GLP-1 Analogues in the Treatment of Obesity-Related Asthma Phenotype

**DOI:** 10.3390/biomedicines13112610

**Published:** 2025-10-24

**Authors:** Joanna Radzik-Zając

**Affiliations:** Department of Allergology and Internal Medicine, Wroclaw Medical University, 50-367 Wroclaw, Poland; joanna.radzik-zajac@umw.edu.pl

**Keywords:** asthma, asthma phenotypes, GLP-1 analogues, GLP-1 receptor, obesity

## Abstract

Asthma and obesity are two common chronic diseases of growing clinical and social importance. One of the recognised phenotypes of asthma is obesity-related asthma, which is characterised by a more severe course, resistance to glucocorticosteroids, increased inflammation and poorer symptom control. This article discusses the complex pathophysiological mechanism of this phenotype, considering the role of chronic inflammation, immune dysregulation and metabolic disorders resulting from obesity. The potential role of glucagon-like peptide-1(GLP-1) receptor analogues as an innovative therapeutic option in the treatment of asthma in obese individuals, both with and without type 2 diabetes mellitus (T2DM), is also analysed. A literature review indicates that glucagon-like peptide-1 receptor analogue (GLP-1RA) drugs, in addition to their hypoglycaemic and weight-reducing effects, also exhibit anti-inflammatory activity in the respiratory system and may reduce the frequency of asthma exacerbations and improve asthma control. The article reviews current data from experimental and clinical studies, emphasising the need for further research on the use of GLP-1RA as an adjunct to conventional asthma therapy in the context of the obese asthma phenotype.

## 1. Introduction

Asthma is a chronic disease of global significance. Its prevalence was estimated to be 262 million cases in 2019 [1]. The definition of asthma states that it is a heterogeneous condition characterised by chronic inflammation of the airways and symptoms caused by bronchial obstruction, such as shortness of breath, wheezing and chest tightness, which vary in severity and frequency. These symptoms are associated with different degrees of restriction of exhaled air flow through the airways, which can and should be measured using a spirometer or peak flow meter [2]. The diversity of asthma, depending on its clinical course, pathophysiological basis, triggering factors, and prevalence in specific social groups, has prompted many researchers to distinguish between different asthma phenotypes [3,4,5,6]. A phenotype is defined as the visible characteristics of an organism resulting from the interaction between its genetic composition and the environment [5,7]. Among asthma phenotypes, obesity-related asthma is distinguished [2,8]. Obesity is the pandemic of the 21st century. In 2022, 2.5 billion adults were overweight, including more than 890 million adults who were obese. Over 390 million children and adolescents aged 5 to 19 were overweight in 2022. The prevalence of overweight (including obesity) among children and adolescents aged 5 to 19 increased dramatically from just 8% in 1990 to 20% in 2022 [9]. The World Obesity Federation Atlas, published on World Obesity Day, 4 March 2025, predicts that the total number of adults living with obesity will increase by more than 115% between 2010 and 2030, from 524 million to 1.13 billion [10]. Scientific data show that both obesity and asthma are serious global health problems. Therefore, scientists are very interested in methods of treating obesity and optimising and personalising asthma treatment, especially regarding its Th2-high or Th2-low endotypes and phenotypes. The phenotype of obese asthma, with its typical characteristics such as resistance to glucocorticosteroids, more severe course, higher risk of hospitalisation, impaired respiratory mechanics, and increased systemic inflammation [11,12,13,14], is a particular challenge for clinicians. Research indicates that GLP-1 receptor analogues may be an important adjunctive treatment for patients with asthma and obesity/T2DM. It has been suggested that using GLP-1RAs in adult patients with asthma and T2DM reduces the expression of key inflammatory pathways in the airways, thereby reducing the risk of asthma exacerbations, regardless of improvements in glycemic control or weight loss [15].

This article discusses the distinguishing features of the obesity-related asthma phenotype and explores the potential benefits of adding GLP-1 analogues to conventional asthma therapy in patients with or without T2DM.

## 2. Methodology

Electronic databases were searched for original articles, meta-analyses, and systematic reviews addressing the obesity-related asthma phenotype and the potential role of GLP-1 analogues in asthma treatment among obese individuals, including their impact on asthma control and their anti-inflammatory effects on the respiratory system. The literature search was conducted in PubMed and Google Scholar between 1 January 2011 and 23 September 2025. The starting point of 2011 was chosen because from that time onward an increasing number of studies began to report beneficial effects of GLP-1 analogues on diseases other than T2DM, as well as their anti-inflammatory and respiratory effects. The following keywords were used, both individually and in combination: “asthma,” “obesity,” “GLP-1,” and “GLP-1 receptor.” Reference lists of the retrieved articles were also screened for additional relevant studies. Only articles published in English were included; studies published in other languages were excluded.

## 3. Asthma Phenotypes and Endotypes

The need for personalised asthma therapy has prompted researchers to more precisely classify asthma into phenotypes and endotypes. A phenotype is defined as comprising the clinical features of asthma, including age of onset, triggers, comorbidities, treatment response, and disease evolution over time. However, an asthma endotype is more difficult to define, as it involves underlying immunopathological mechanisms and requires reliable biomarkers to guide the implementation of tailored treatment [16,17]. Asthma patients can currently be divided based on the contribution of the inflammatory response to Th2-high and Th2-low endotypes. Within these groups, phenotypes can be distinguished based on additional features such as allergy status, age of onset, inflammatory parameters (i.e., high levels of eosinophils in blood or sputum), comorbidities (e.g., nasal polyposis), and response to inhaled corticosteroids [18,19]. The Th-2-high asthma endotype is well-defined. Th2-high asthma is defined as allergic asthma involving an immune-inflammatory response stimulated by Th-2 cells, which secrete primarily the prototypical cytokines IL-4, IL-5, and IL-13 and stimulate type 2 immunity [20]. Airway type 2 immune responses are mediated primarily by eosinophils, mast cells, basophils, Th2 cells, type 2 innate lymphoid cells (ILC2), and IgE-producing B lymphocytes. Upon entering the airway epithelium, inhaled allergens can activate Toll-like receptors (TLRs), a class of pattern recognition receptors involved in innate immunity. TLR activation induces epithelial synthesis of innate cytokines, such as thymic stromal lymphopoietin (TSLP), IL-25, and IL-33, which can induce the development of an adaptive Th2 response [21]. No specific clinically relevant biomarkers have been identified for Th2-low asthma. However, it is known that neutrophilic inflammation in the airways is induced in Th2-low asthma, typically associated with more severe asthma phenotypes. A specific lineage of CD4+ effector T lymphocytes, secreting IL-17 and hence termed Th17, appears to play a key role in airway neutrophilia [21]. In Th2-low asthma, the high blood and sputum eosinophilia and elevated Fractional Exhaled Nitric Oxide (FeNO) levels typical of Th2-high asthma are absent. Th2-low asthma is associated with a later age of onset, a female predominance, the need for high doses of glucocorticosteroids, and obesity [22] (Table 1). High doses of glucocorticosteroids lead to exacerbation of metabolic disorders and obesity. Considering the complex pathogenic mechanisms in obesity-related asthma, Jutel et al., classified obesity-related asthma as a tissue-dependent hypersensitivity reaction, type VI hypersensitivity reaction, related to metabolic dysregulation [8].

According to current recommendations, the primary medications used in asthma are inhaled bronchodilators and glucocorticosteroids [2]. Additionally, monoclonal antibodies directed against IgE or type 2 cytokines interleukin (IL)-4, IL-5, and IL-13 are used, which have been shown to be highly effective in alleviating exacerbations and symptoms in people with severe allergic and eosinophilic asthma [23]. However, 30–50% of patients with severe non-allergic, non-eosinophilic asthma with Th2-low inflammation do not derive adequate benefit from this therapy [23]. The treatment of patients with Th2-low asthma, including obesity-related asthma, is a significant clinical problem.

## 4. The Impact of Obesity on Asthma

According to the WHO definition, obesity is a chronic, complex disease characterised by excessive fat accumulation, which can negatively impact health. Obesity can lead to an increased risk of T2DM, heart disease, poor bone health, poor reproductive health, and an increased risk of certain cancers. In adults, obesity is defined as a body mass index (BMI) of 30 kg/m^2^ or greater [9]. Obesity is not only a risk factor for the development of asthma, but also significantly influences the clinical course of asthma with characteristic features for a distinct phenotype of this disease in both children and adults [24]. Obese asthmatics have a more severe course of asthma, poorer control, a higher risk of hospitalisation, and a poorer quality of life compared to asthmatics with a normal body weight [25]. Moreover, obese asthmatics have a poorer response to treatment with inhaled glucocorticosteroids [11]. Obese asthma is more common in adult women [12]. Obesity may influence asthma by altering chest wall dynamics, reducing lung volume, and reducing lung compliance [13]. Obesity has been shown to be associated with chronic, low-grade inflammation, known as meta-inflammation (inflammation in metabolic tissues), mediated by macrophages. This inflammation is characterised by a modest increase in circulating pro-inflammatory factors and a lack of clinical signs of inflammation (hence the term “subclinical inflammation”) [14].

Adipose tissue is divided into two types: white adipose tissue (WAT) and brown adipose tissue. WAT is an endocrine organ, as exemplified by the fact that many adipokines, cytokines, and chemokines are released into the bloodstream from white adipose tissue [26]. WAT is composed primarily of adipocytes but also contains preadipocytes, immune cells, fibroblasts, and vascular cells, which are collectively referred to as the vascular-stromal fraction. The number and phenotype of WAT cells differ between obese and lean individuals [27]. Macrophages are the predominant immune cell type found in WAT and appear to be crucial in the development of metainflammation. The number, location, and phenotype of macrophages undergo significant changes in obesity. Although they constitute 10–15% of the WAT cell population in non-obese individuals, their numbers are significantly increased, reaching as much as 40–50% of the WAT cell population in obese humans and mice [26]. The development of inflammation in WAT occurs as a result of hypertrophy and hyperplasia of adipocytes [28]. In turn, the expansion of WAT may lead to hypoxia and subsequent death of adipocyte cells [29]. There are two types of macrophages present in adipose tissue: M1 (classically activated) and M2 (alternatively activated). Studies have shown that in humans and non-obese mice, M2 predominates, secreting the anti-inflammatory interleukins IL-10, IL-1, and the receptor antagonist IL-1Ra. IL-4 and IL-13 are important cytokines for maintaining M2. IL-4 is primarily derived from eosinophils residing in adipose tissue, while IL-13 is derived from type 2 innate lymphoid cells [26]. Arginase production is increased in M2-polarised macrophages. This enzyme blocks the activity of inducible nitric oxide synthase (iNOS) through a number of mechanisms, including competition for the substrate arginine, which is essential for nitric oxide (NO) production. This mechanism causes M2 macrophages to block the inflammatory response and promote oxidative metabolism [28]. In the state of obesity, macrophages are polarised into pro-inflammatory M1, nitric oxide synthase 2 (NOS2) is activated, and reactive oxygen species such as NO, CD11c are produced, as well as tumor necrosis factor alpha (TNF-α), IL-6, IL-1β, IL-12, and monocyte chemotactic protein are secreted [26,30]. Pro-inflammatory mediators such as lipopolysaccharide (LPS) and interferon-gamma (IFN-γ) activate M1 macrophages [31]. In response to adipocyte death, proinflammatory M1 macrophages surround dead and dying cells and remove debris from the damaged area. M1 macrophages are known to stimulate proinflammatory factors and induce insulin resistance [28]. The M1 macrophage pool is maintained by IFN-γ and Toll-like receptor (TLR) ligands, the sources of which are mainly Th1 and CD8+ T lymphocytes, as well as necrotic adipocytes [26]. It has been suggested that asthma and obesity are linked by factors such as chronic inflammation, mitochondrial dysfunction, Th17-induced neutrophilia, macrophage dysregulation, hormonal changes, lipid metabolism, insulin resistance, and altered respiratory mechanics [32]. Other abnormalities seen in obesity include accelerated formation of advanced glycation end products (AGEs), subsequent activation of their receptor, and changes in arginine metabolism, which may play a role in the pathogenesis of asthma and may be modulated by the anti-inflammatory incretin GLP-1. The GLP-1 pathway may be crucial for mitigating this inflammation in asthma [32]. Obesity, T2DM, and lipid disorders are accompanied by increased production of AGEs and subsequent activation of their receptor (RAGE), which are highly reactive, non-enzymatically glycated proteins or lipids involved in modulating the inflammatory response. AGEs can also be ingested from foods prepared at high temperatures (e.g., baked or fried). Interactions between AGEs and their receptor (RAGE) generate oxidative stress and perpetuate inflammatory, thrombogenic, and fibrotic responses [33] (Figure 1).

Overexpression of RAGE or its ligands leads to a pro-inflammatory cascade, activating NF-κB, TNF-α, IL-1β, and IL-8, and is observed in inflammatory and neurodegenerative diseases [32].

Due to the complex pathophysiology of this asthma phenotype, effective treatment of asthma associated with obesity remains a challenge for modern medicine. Current expert recommendations for a stepwise approach to asthma management [2] should be strictly adhered to in asthma treatment. However, in addition to standard asthma therapy, measures aimed at normalising body weight should be implemented, including lifestyle changes. Studies show that even a 10% weight loss associated with lifestyle changes and physical exercise is associated with improved asthma control, improved quality of life, and improved overall well-being [34,35] as well as a reduction in inflammatory parameters, a reduction in the concentration of proinflammatory cytokines (CCL2, IL-4, IL-6, TNF-α) and leptin, an improvement in lung function forced expiratory volume in 1 s (FEV1), forced vital capacity (FVC) and expiratory reserve volume (ERV), and an increase in the concentration of vitamin 25(OH)D, anti-inflammatory cytokine IL-10 and adiponectin [36].

## 5. The Role of GLP-1 Receptor Analogues in the Treatment of Asthma in Obese Individuals

In recent years, considerable attention has been devoted to research on the GLP-1 receptor (GLP-1R). The GLP-1R is present on the surface of various cells in the human body. This receptor has significant therapeutic potential in T2DM, metabolic syndrome, and obesity. Activation of the GLP-1R increases insulin secretion, inhibits glucagon release, delays gastric emptying, and reduces food intake through central appetite suppression [37].

However, the scope of action of GLP-1 RA is much broader, and studies show their possible use in the treatment of diseases such as inflammation of the musculoskeletal system, cardiovascular diseases, kidney diseases, non-alcoholic fatty liver disease (NAFLD, now named MASLD), neurodegenerative diseases and various cancers [38,39,40,41,42]. Studies suggest that GLP-1R agonists reduce airway inflammation through a number of mechanisms, including increased surfactant production, decreased mucus secretion, decreased type 2 inflammatory signalling, and increased smooth muscle relaxation [15]. It’s important to note that GLP-1 and glucose-dependent insulinotropic polypeptide (GIP) are two naturally occurring hormonal peptides produced in the gastrointestinal tract, known as incretins. Together, they are responsible for a key hormonal regulation known as the incretin effect, which causes insulin secretion following oral glucose administration to be two to three times higher than following isocaloric intravenous glucose administration [43].

GLP-1 is a 30- to 31-amino acid incretin produced by posttranslational processing of proglucagon. This glucose-lowering hormone is secreted by intestinal enteroendocrine L cells in response to nutritional and inflammatory stimuli and by neurons in the nucleus of the solitary tract in the brainstem. GLP-1 activates the GLP-1R, which is coupled to seven transmembrane G proteins. GLP-1 receptors are present on pancreatic β-cells, lung epithelial cells, atrial cardiomyocytes, vagal afferents, neurons in many brain regions, and cells lining the gastric pits and the mucosa of the small intestine. The GLP-1R can bind to Gs or Gq proteins, leading to an increase in intracellular cAMP and/or Ca^2+^ concentration and activation of the protein kinase C (PKA), Epac-2, phospholipase C, and extracellular signal-regulated kinase (ERK1/2) signalling pathways. The hypoglycemic activity of GLP-1 is associated with stimulation of glucose-dependent insulin secretion, inhibition of glucagon production, and regulation of pancreatic islet cell proliferation, differentiation, and survival. Under physiological conditions, GLP-1 is rapidly degraded by dipeptidyl peptidase-4 (DPP-4) after release [44].

Recently, many studies on in vivo/ex vivo animal models have shown that, in addition to the basic action of counteracting hyperglycemia and reducing body weight, GLP-1R analogues also reduce inflammation in the airways, reduce mucus secretion, inhibit lung fibrosis, and induce bronchodilation [45,46,47,48,49]. Studies on the effects of GLP-1 analogues in asthma patients are very limited. A pilot study in a small group of asthma patients followed for one year showed that asthma exacerbations occurred shortly after liraglutide treatment was discontinued, whereas none of the patients taking liraglutide experienced clinical worsening of asthma [50]. Retrospective studies suggest a positive effect of GLP-1 analogues in controlling asthma/chronic lower respiratory tract disease. In patients with chronic lower respiratory tract disease and T2DM, fewer disease exacerbations were observed in patients taking GLP-1 analogues compared to patients taking other antidiabetic medications [51,52]. Moreover, the use of GLP-1 analogues in patients with T2DM and asthma has been shown to reduce the systemic concentration of periostin, which is a biomarker of airway eosinophilia [53].

GLP-1R belongs to a family of proteins known as Gs protein-coupled receptors, which activate metabolic pathways such as the cyclic adenosine monophosphate (cAMP)/PKA, cAMP/guanine nucleotide exchange factor, and phosphatidylinositol-3/protein kinase C (PKC) pathways. Activation of cAMP is the mechanism by which β2-adrenergic receptor agonists, prostaglandin E2, and phosphodiesterase inhibitors cause airway smooth muscle relaxation [54]. The bronchodilatory efficacy of the GLP-1R agonist was demonstrated in an experimental ex vivo model of human isolated bronchi [49].

GLP-1R signalling was examined in a polygenic model of obesity using TALLYHO (obese) mice. TALLYHO mice had greater allergen-induced airway neutrophilia and expression of the pulmonary proteins IL-5, IL-13, CCL11, CXCL1, and CXCL5, in addition to ICAM-1 expression on lung epithelial cells, compared with lean mice. Allergen exposure increased IL-33 levels in bronchoalveolar lavage fluid (BALF) of TALLYHO (obese) and SWR (lean) mice compared with exposure to phosphate-buffered saline (PBS), but there was no difference in IL-33 levels in BALF between TALLYHO strains and lean controls. However, TALLYHO mice, but not lean mice, had significantly higher levels of thymic stromal lymphopoietin (TSLP) in the airway BALF after allergen exposure compared with PBS exposure. Treatment with a GLP-1R agonist significantly reduced allergen-induced TSLP and IL-33 release in TALLYHO mice but did not reduce airway neutrophil counts in TALLYHO mice. These results suggest that GLP-1R agonist treatment may be a novel pharmacological strategy for the treatment of obese individuals with asthma by inhibiting aeroallergen-induced neutrophilia [47].

In human studies, the functional effect of the GLP-1 analogue on human eosinophils was demonstrated by attenuating LPS-stimulated surface expression of CD69, CD11b and the production of IL-4, IL-8 and IL-13 [55]. GLP-1R agonists may have an independent lung protective role even in individuals without lung disease, as evidenced by the relative increase in mean FEV1 and FVC in individuals taking metformin and GLP-1R agonists compared with metformin alone or metformin and insulin. In adult patients with T2DM treated with GLP-1R agonists, FEV1 increased from baseline after 3 months of treatment, and the improvement in FEV1 was maintained through 24 months of follow-up (mean increase in FEV1 was 195 mL; maximum increase in FEV1 was 218 mL after 24 months of treatment). After 3 months of treatment with GLP-1R agonists, the change from baseline in FVC was 280 mL, and FVC values remained stable through 24 months of follow-up; mean increase in FVC was 253 mL [56].

Studies show that among patients with asthma and T2DM, those treated with GLP-1R agonists experienced fewer asthma exacerbations compared with patients using other antidiabetic medications [52,55,57]. Obese asthmatics (not suffering from T2DM) using GLP-1 analogues have better asthma control, regardless of whether this is related to weight loss, diabetes control, or anti-inflammatory effects of GLP-1 analogue therapy [58]. Another mechanism supporting the treatment of obesity-related asthma is the inhibition of the aberrant arginine gene product, asymmetric dimethylarginine, which is a competitive inhibitor of NO synthase and thus increases NO levels [59].

GLP-1 signalling using GLP-1 analogues is a promising new target for treating chronic airway inflammation in asthma. The multi-organ anti-inflammatory effects of GLP-1 analogues can be exploited in the treatment of individuals with T2DM, asthma, and co-occurring obesity. Currently, GLP-1 analogues such as albiglutide, dulaglutide, liraglutide, semaglutide, and the dual GIP/GLP-1 receptor agonist tirzepatid are clinically used in the treatment of T2DM and obesity. These drugs are approved by the Food and Drug Administration (FDA) [43]. Studies on the effects of selected GLP-1 analogues and the GIP/GLP-1 analogue on the respiratory system are presented below (Table 2).

Lixisenatide—there are no scientific reports on the role of lixisenatide in asthma and asthma in obese people.

Eksenatyd—lack of research in asthma. The study concerned exendin-4—a GLP-1 analogue, isolated from the saliva of the venomous Gila monster lizard (Heloderma suspectum), the synthetic form of which is exenatide [64]. It was noted that apart from the typical effect of GLP-1 analogues, increasing insulin secretion, reducing postprandial glycemia, reducing appetite and delaying gastric emptying, causing weight loss [44], exenatide has anti-inflammatory, neuroprotective and anti-bronchial hyperreactivity effects [49]. In an ex vivo study using isolated human bronchi, Rogliani et al. [49] demonstrated a moderate, time- and dose-dependent bronchodilator effect of exendin-4 on isolated human bronchi, which was independent of the epithelium. The bronchodilator effect was mediated by activation of the cAMP-dependent PKA pathway.

Liraglutide—Hur J. et al. [60] demonstrated that a GLP-1R agonist effectively induced weight loss, inhibited eosinophilic bronchitis, and bronchial hyperresponsiveness (BHR) in obese asthmatic mice. These effects were mediated by suppression of NLRP3 and IL-1β inflammasome activity. Toki et al. [61] challenged the airways of mice with Alternaria alternata extract, an aeroallergen with protease activity that is associated with severe asthma exacerbations. GLP-1R agonist liraglutide, initiated 2 days before the first Alternaria extract challenge, inhibited the expression and release of IL-33 in the lung or into bronchoalveolar fluid and further inhibited IL-33 release in the lung in response to aeroallergen challenge. GLP-1R agonist initiated 2 days before the first Alternaria extract challenge inhibited IL-5 and IL-13 production by lung ILC2s, reduced mucus and airway reactivity, and reduced pulmonary eosinophilia. Gou et al. [48] demonstrated that liraglutide attenuated bleomycin-induced pulmonary fibrosis in mice. Ovalbumin (OVA)-induced asthma was effectively alleviated by liraglutide, with decreased airway inflammation and mucus hypersecretion. The mechanism of this effect was likely PKA-dependent inactivation of NF-κB in mice [45].

Semaglutide—ameliorated acute lung injury (ALI) by blocking the histone deacetylase 5/nuclear factor kappa-light-chain-enhancer of activated B cells (HDAC5/NF-κB) pathway. The study established an in vivo ALI model based on Sprague-Dawley (SD) rats and an in vitro ALI model based on human pulmonary artery endothelial cells (HPAEC) with LPS. It was found that semaglutide could play an ameliorative role in LPS-induced ALI in rats. Furthermore, it was also revealed that it could ameliorate LPS-induced HPAEC cell damage by inactivating the HDAC5-mediated NF-κB signalling pathway [62].

Tirzepatide, a dual agonist of the gastric inhibitory peptide receptor (GIPR) and the GLP-1R, reduces serum leptin levels. Because leptin increases the activation of Th2 lymphocytes and ILC2, exacerbating type 2 inflammation, lowering leptin levels with tirzepatide treatment may be one mechanism for reducing Th2 lymphocyte activation, which in turn leads to inhibition of allergic lung inflammation. In a mouse model of aeroallergen-induced asthma in obese individuals, tirzepatide exhibits antiallergic and anti-inflammatory effects [63] (Table 2).

## 6. Conclusions

GLP-1 analogues, when used as adjunctive therapy for asthma control in patients with comorbid obesity or T2DM, may represent a breakthrough in the treatment of this patient population. Due to their complex mechanism of action and well-documented anti-inflammatory effects, GLP-1 analogues may constitute an important component of personalised therapy for the obesity-related asthma phenotype, which poses a therapeutic challenge for clinicians. Further randomised controlled trials in humans are needed to confirm the benefits of incorporating GLP-1 analogues into asthma management in obese individuals. Such RCTs should include priority endpoints such as asthma control and exacerbation rates assessed by validated questionnaires (ACT, AQLQ, mini-AQLQ, ACQ), inflammatory asthma biomarkers (total IgE, FeNO, eosinophil count), and lung function parameters (PEF, FEV_1_, FVC, FEV_1_/FVC). The results of such studies will provide clinicians and researchers with essential information for establishing standards of care for obesity-related asthma.

## Figures and Tables

**Figure 1 biomedicines-13-02610-f001:**
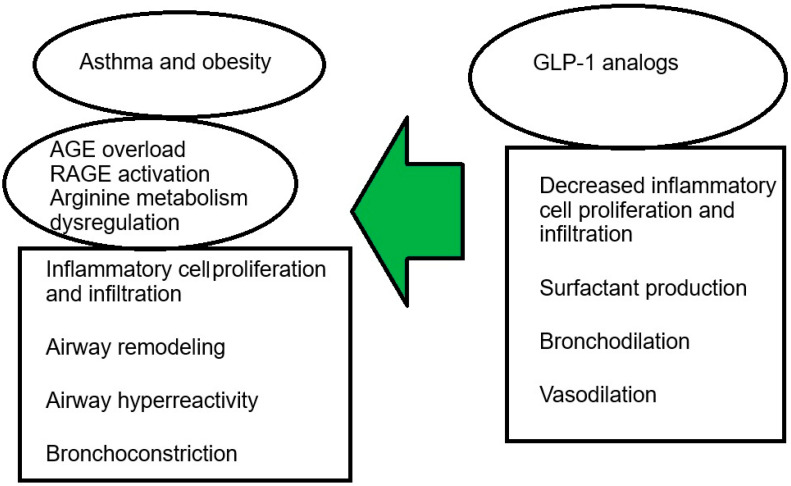
The effect of GLP-1 analogues in asthma [33].

**Table 1 biomedicines-13-02610-t001:** Characteristics distinguishing asthma endotypes Th2–high and Th2–low [20].

Asthma Th2-High	Asthma Th2-Low
Children and adults	Adults (female predominance)
Eosinophilic inflammation	Neutrophilic inflammation
Th2 inflammatory cytokines: IL-4, IL-5, IL-13	Th1 inflammatory cytokines: IL-8, IL-17
Responsiveness to glucocorticosteroids	Lack of responsiveness to glucocorticosteroids
Responsiveness to inhibitors of type 2 inflammation	Lack of responsiveness to inhibitors of type 2 inflammation

**Table 2 biomedicines-13-02610-t002:** The effect of selected GLP-1 analogues on the respiratory system.

GLP-1 Analogues	In Vivo/Ex Vivo Animal Studies	Human Studies
Lixisenatide	No data available	No data available
Exenatide /exendin-4	Isolated human bronchi—dilation [49]	No data available
Liraglutide	Inhibition of eosinophilic bronchitis and bronchial hyperresponsiveness in mice [60] Inhibition of IL-33, IL-5, IL-13 release, reduction in pulmonary eosinophilia [61] Reduction in pulmonary fibrosis [48] Reduction in inflammation and mucus secretion in OVA-induced asthma [45]	No data available
Semaglutide	Reduces acute lung injury [62]	No data available
**GIP/GLP-1R agonist**		
Tirzepatide	Reduction in leptin concentration, anti-inflammatory and anti-allergic effects in asthma in mice [63]	No data available

## Data Availability

No new data were created.

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
