# Peer review of "The Role of GLP-1 Analogues in the Treatment of Obesity-Related Asthma Phenotype"

_biomedicines, 2025, doi:10.3390/biomedicines13112610_

Round 1

Reviewer 1 Report

Comments and Suggestions for Authors

General comments

As the title of this review suggests, the information on asthma treatment with GLP-1 analogs is limited and contains many basic descriptions. It is better to be reduced the basic descriptions on asthma, GLP-1, inflammation, etc. and enrich the content indicated by the title of this paper. However, this review will provide useful insights into the treatment of obesity-related asthma.

Individual comments

1, L65-67: Briefly explain why authors limited search range to 2011 or later.

2, Figure 1: No explanation for asterisks (*, **).

3, Figure 1: The calculations in the right box (removed papers) and the left box (remaining papers) do not match.

4, The abbreviations frequently used in the manuscript, “DMt2” (L219, 289, 290, 320, 322, 330, etc.) are not included in the abbreviation list (L388), and the common abbreviation for “DMt2” is "T2DM."

5, L246: "the GLP-1 receptor and its analogues" is an inaccurate statement. Please rephrase.

  L248: Authors should remove "and its" from the phrase "This receptor and its agonists"

6, L254: The name "NAFLD" has been changed to "MASLD," including the full spelling. While "NAFLD" likely refers to the original text, the change in name should be clearly stated in this review paper.

7, L259: The correct full spelling "glucose-dependent insulinotropic polypeptide (GIP)"is used here, but the abbreviation list (L391) uses the old full name. Correct the list.

8, L297: Insert “receptor” after “GLP-1”.

9, L317: About “FEV1/FVC”, add brief explanation.

10, Table 2: Table 2 is difficult to understand. For example, is the reference (60) added to the top line (heading) of the last column in the wrong position? The meaning of this last column is unclear. Why are not all the items filled in?

11, L355-361: Remove repeated phrases “Treatment with the …”.

12, L367-369: This sentence is probably incorrect. It seems that "in vitro" and "in vivo" have been interchanged.

Comments on the Quality of English Language

No comments because this reviewer is not English native.

Author Response

                      Please read the attachment.

Reviewer 2 Report

Comments and Suggestions for Authors

The manuscript presents a timely and well-structured review addressing the potential role of Glucagon-like Peptide-1 Receptor Agonists (GLP-1RAs) in managing the obesity-related asthma phenotype. The topic sits at the intersection of two significant public health challenges and explores a promising avenue for personalized therapy. In its current form, the manuscript is close to being suitable for publication but would benefit from minor revisions to enhance its clarity and completeness.

Comments and Recommendations for Improvement:

1) The manuscript is currently positioned as a systematic review; however, its methodology aligns more closely with a narrative review. It is advised to reclassify the work as a Narrative Review to meet journal standards. Alternatively, the methodology section could be expanded with a detailed search strategy and study evaluation criteria.
2) The conclusion could be strengthened by specifying the prospects for future research—for instance, by outlining priority endpoints for future RCTs (e.g., exacerbation rate, inflammatory markers, lung function parameters).

Round 2

Reviewer 1 Report

Comments and Suggestions for Authors

The revised manuscript largely addresses the individual comments, however, there appears to be no response to the general comments on the original manuscript.